# Challenges in Diagnosis and Prevention of Iatrogenic Endometriosis as a Long-Term Surgical Complication after C-Section

**DOI:** 10.3390/ijerph19052791

**Published:** 2022-02-27

**Authors:** Radu Neamtu, George Dahma, Adelina Geanina Mocanu, Elena Bernad, Carmen-Ioana Silaghi, Lavinia Stelea, Cosmin Citu, Amadeus Dobrescu, Felix Bratosin, Mirela Loredana Grigoras, Andrei Motoc, Sorin Dema, Marius Craina, Veronica Daniela Chiriac, Adrian Gluhovschi

**Affiliations:** 1Discipline of Obstetrics and Gynecology, “Victor Babes” University of Medicine and Pharmacy Timisoara, 300041 Timisoara, Romania; radu.neamtu@umft.ro (R.N.); george_dahma@yahoo.com (G.D.); adelinaerimescu@yahoo.com (A.G.M.); ebernad@yahoo.com (E.B.); silaghi.carmen@gmail.com (C.-I.S.); stelea_lavinia@yahoo.com (L.S.); citu.ioan@umft.ro (C.C.); crainamariuslucian@gmail.com (M.C.); chiriac.veronica@umft.ro (V.D.C.); adigluhovschi@yahoo.com (A.G.); 2Discipline of General Surgery, “Victor Babes” University of Medicine and Pharmacy Timisoara, 300041 Timisoara, Romania; dobrescu.amadeus@umft.ro (A.D.); felix.bratosin7@gmail.com (F.B.); 3Discipline of Anatomy and Embryology, “Victor Babes” University of Medicine and Pharmacy Timisoara, 300041 Timisoara, Romania; amotoc@umft.ro; 4Discipline of Radiology, “Victor Babes” University of Medicine and Pharmacy Timisoara, 300041 Timisoara, Romania; sorindema@yahoo.com

**Keywords:** endometriosis, iatrogenic disease, surgical scars, cesarean section, amniotic fluid

## Abstract

Endometriosis is a gynecological condition caused by the development of endometrial tissue outside the uterine cavity. Naturally, it commonly occurs at locations such as the ovaries and pelvic peritoneum. However, ectopic endometrial tissue may be discovered outside of the typical sites, suggesting the possibility of iatrogenic seeding after gynecological operations. Based on this hypothesis, we developed a study aiming to establish the root cause of atypical implantation of endometrial foci, as the main end point, and to determine diagnostic features and risk factors for this condition, as a secondary target. The research followed a retrospective design, including a total of 126 patients with endometriosis who met the inclusion criteria. A group of 71 patients with a history of c-section was compared with a control group of patients with endometriosis and no history of c-section. Endometriosis that developed inside or in close proximity to surgical incisions of asymptomatic patients before surgical intervention was defined as iatrogenic endometriosis. Compared with patients who did not have a c-section, the c-section group had significantly more minimally invasive pelvic procedures and multiple adhesions and endometriosis foci at intraoperative look (52.1% vs. 34.5%, respectively 52.1% vs. 29.1%). The most common location for endometriosis lesions in patients with prior c-section was the abdominal wall (42.2% vs. 5.4%), although the size of foci was significantly smaller by size and weight (32.2 mm vs. 34.8 mm, respectively 48.6 g vs. 53.1 g). The abdominal wall endometriosis was significantly associated with minimally invasive pelvic procedures (correlation coefficient = 0.469, *p*-value = 0.001) and c-section (correlation coefficient = 0.523, *p*-value = 0.001). A multivariate regression analysis identified prior c-section as an independent risk factor for abdominal wall endometriosis (OR = 1.85, *p*-value < 0.001). We advocate for strict protocols to be implemented and followed during c-section and minimally invasive procedures involving the pelvic region to ensure minimum spillage of endometrial cells. Further research should be developed to determine the method of abdominal and surgical site irrigation that can significantly reduce the risk of implantation of viable endometrial cells. Understanding all details of iatrogenic endometriosis will lead to the development of non-invasive disease diagnosis and minimally invasive procedures that have the potential to reduce postoperative complications.

## 1. Introduction

Iatrogenic endometriosis (IE) is defined by the appearance of endometrial glands and stroma outside the uterus following certain surgical procedures, including complete or supracervical hysterectomy, myomectomy, cesarean section, and the endometrial tissue seeding of surgical scars during these operations [1]. Cesarean scars such as skin and uterine scars, trocar insertion sites, sigmoid colon, ovaries, bladder, vaginal vault, and parietal peritoneum are the most prevalent locations for IE [2,3]. The transvaginal ultrasound examination is the first step in detecting endometriosis, since it is readily accessible, non-invasive, and reasonably inexpensive [4]. It is a precise technique for ovarian endometrial cysts but has a sensitivity of roughly 40% for endometriosis occurring outside the ovary [5,6]. With a sensitivity of about 95%, magnetic resonance imaging (MRI) enables a more precise diagnosis when ultrasound results do not match the clinical picture of the patient. This technique is especially beneficial for diagnosing retroperitoneal foci located in a variety of locations, including the rectovaginal fascia or adenomyosis. It is, however, ineffective for identifying peritoneal endometriosis and foci less than 3 mm in diameter [7]. Computed tomography (CT) is less effective in diagnosing pelvic endometriosis due to the presence of bone structures. The lesions that are characteristic of this illness lack particular characteristics that may be recognized by MRI or CT [8]. Biochemical diagnostics may involve the determination of the levels of CA125, a highly sensitive but imprecise marker [9]. Laparoscopy or laparotomy is the gold standard for diagnosing pelvic endometriosis [10], since the operation enables the assessment of the uterus, appendages, peritoneum, adhesions, and the size and number of foci, hence determining the degree of endometriosis.

There is scarce evidence that iatrogenic endometriosis behaves or develops differently from non-iatrogenic endometriosis, since the general clinical presentation and patient complaints with endometriosis mostly include dysmenorrhea, dysuria, dyschezia, chronic pelvic pain, and infertility [11]. These factors serve as indicators for treatment [12]; thus, the care of iatrogenic endometriosis can significantly differ from that of non-iatrogenic endometriosis by the main medical or surgical approach. The same approach should be explored for symptom management and surgical treatment for endometriotic lesions excision. The existing evidence describes patients with symptom onset of iatrogenic endometriosis after c-section occurring anytime between the first menstruation after surgery and seven years following surgery, while after hysterectomy, IE is reported to occur at an incidence of 1.4 percent [13,14].

Although endometriosis is believed to develop to cancer in less than 1% of instances, investigations have established it as a precursor to clear cell and endometrioid ovarian carcinomas [15]. Women with endometriosis have a two- to four-fold increased chance of acquiring these tumors; in addition, mutational investigations have shown a clonal link between endometriosis-associated ovarian carcinomas and endometriotic lesions [16]. Nevertheless, the slightest increase in the risk of malignant transformation of this preventable pathology should raise awareness during obstetrical and gynecological interventions.

The available data regarding iatrogenic IE after surgical intervention were sparse and were derived from case reports and short case series, but this topic has recently sparked more interest. The majority of research includes instances of gynecologic surgery for benign conditions such as adenomyosis, uterine leiomyomas, and fibroids [17,18], with some of them raising the hypothesis of post-cesarean scar endometriosis [19]. Therefore, in the present study, we attempted to assess imaging and histopathological findings from patients treated in our clinic in conjunction with their clinical presentation to determine the mechanism of endometrial cells spread from their physiological site in atypical locations such as the abdominal wall. A second objective was to determine diagnostic features and risk factors for this condition to allow future implementation of proper strategies in preventing this long-term complication after surgical procedures conducted in the obstetrics and gynecology departments.

## 2. Materials and Methods

This research follows an observational retrospective design using the database of the tertiary hospital “University Clinic of Obstetrics and Gynecology Bega” from Timisoara, Romania. The project was authorized by the institution’s ethical review committee in November 2021, with permission number A59. Between January 2010 and November 2021, this study enrolled a group of patients admitted to our clinic with a diagnosis of endometriosis who underwent a cesarean section and a control group of patients with endometriosis and no history of pelvic surgical interventions. All patients presented at our clinic with chronic pelvic pain at follow-up or in subsequent hospital presentations. The following criteria were used to determine inclusion in the first study group: (1) the patient had undergone at least one cesarean section; (2) the patient had no complains related to endometriosis before surgery; (3) no macroscopic findings of endometriosis were observed or documented by the gynecologist while the cesarean section was being performed; (4) the patient developed endometriosis symptoms following the cesarean delivery; (5) the endometrial foci were surgically excised; and (6) the histopathological diagnosis for each excised lesion was endometriosis. A total of 71 patients satisfied the aforementioned criteria and were included in the group of cases with a history of surgery, while 55 patients met the inclusion requirements for endometriosis with no prior history of surgical interventions. 

All patients’ baseline characteristics and surgical procedures were documented in the hospital database in addition to paper patient records that were examined by qualified professionals participating in the present research. We used a computerized database search to determine the exact diagnosis codified by the International Classification of Diseases (ICD-10), and procedures codified by the Current Procedural Terminology (CPT). The clinical features characterizing IE that had been preliminarily filtered from the database included a palpable mass under or distant from the scar and cyclic discomfort and swelling during menstruation. The latency period was defined as the time interval between the commencement of the cesarean section and the start of IE symptoms. Patient age, age at IE diagnosis, parity, abortion history, delivery history, incision type, symptoms, mass size, latency period, the time between symptoms and surgery, ultrasound assessment, radiologic imaging, operative findings, and histological evaluations were all retrieved as study variables.

After establishing a presumptive diagnosis by clinical presentation and imaging studies, the patients underwent pelvic laparoscopic surgical exploration or, in some cases, laparotomy, with excision and biopsy of the respective tissues, according to hospital protocols in place at the time. The analysis of the histologically processed tissue specimens was carried out as a single-center study, and tissue sampling was performed according to existing guidelines [20]. Tissue samples were initially fixed in 10% buffered formalin and then photographed with a graduated ruler for measurement. After paraffin embedding, staining with hematoxylin-eosin was used to determine the major characteristics of endometrial tissue, as it is the most common method of diagnosis in surgical pathology [21].

Data analysis was performed using the SPSS statistical software version 26.0 and MedCalc v.19. The Student’s *t*-test was used to compare normally distributed, continuous data, while the Mann–Whitney *U*-test helped assess non-normally distributed, continuous data; otherwise, the Kruskal–Wallis test was employed. The Chi-square and Fisher exact tests were used to assess the proportions of categorical data. Pearson’s and Spearman’s correlations were used to determine the relationship between parametric and non-parametric data, respectively. A multivariate regression analysis was performed to determine independent risk factors associated with abdominal wall endometriosis. The significance threshold was set for alpha = 0.05.

## 3. Results

Table 1 describes the general characteristics of the 71 patients included in the study who were diagnosed with endometriosis following a cesarean section. The average age of females diagnosed with endometriosis was 33 years old, with a surgical intervention at average at 28 years old, a latency of 27.5 months from the intervention until the diagnosis of endometriosis, and an average of more than 30 months from diagnosis until seeking medical treatment. Most of the patients (59%) had a single cesarean delivery, with the vast majority being incised in a Pfannenstiel approach (87.3%). The endometriomas found in all patients included in our study were generally multiple in number in more than 80% of cases, while 42.2% of them were localized at the incision site, which were more likely to be found in the fascia (56.3%) and muscular layer (21.1%). Almost all patients presented to medical evaluation presenting cyclic pain in the pelvic and mid-abdominal region (87.3%), while the second and third most common complaints were dysmenorrhea (69.1%) and the presence of an abdominal mass (61.9%).

By comparing the group of patients with a positive history of cesarean section with the patients without previous c-sections, we observed a significant difference in the minimally invasive procedures, where the first group of patients had significantly more interventions (52.1% vs. 34.5%, *p*-value = 0.049). The size and weight of endometrial foci were significantly larger in patients without a history of c-sections, with an average of 34.8 mm in size and 53.1 g in weight. The most prevalent position of endometriosis in the group of 71 patients with c-sections was the abdominal wall (42.2%), while in the comparison group, it involved mostly the ovaries in 50.9% of cases. The differences were highly significant. The intraoperative look accounted for multiple differences between groups, where 80% of patients with no history of c-sections did not have adhesions, compared with 53.6% in the other group (*p*-value = 0.002). Multiple adhesions and endometriosis foci were discovered intraoperatively in patients with previous c-sections (52.1%), while only 29.1% had multiple lesions in the group of patients without a history of c-section (*p*-value = 0.017) (Table 2).

The imaging studies described below present representative findings for the patients described above. The initial suspicion of endometriosis was approached by abdominal ultrasound, as seen in Figure 1A,B. Further evaluation required an MRI study (Figure 2), after which the patients underwent elective surgery to excise the masses and perform a biopsy, as shown in Figure 3A,B.

The correlation analysis determined that there was a significant positive association between minimally invasive pelvic procedures and abdominal wall location of the endometriosis (rho value = 0.469, *p*-value = 0.001) and the number of endometriosis foci (rho value = 0.465, *p*-value = 0.001), while the correlation with foci weight and size was significantly negative (Table 3). Additionally, we observed a significant positive association between c-section and abdominal wall location of the endometriosis (rho value = 0.523, *p*-value = 0.001) and the number of endometriosis foci (rho value = 0.488, *p*-value = 0.001), while the correlation with foci size was significantly negative (rho value = −0.258, *p*-value = 0.003).

The risk factor analysis determined that the number of pregnancies (OR = 1.32, CI (1.01–1.64), *p* = 0.038) and the cesarean section (OR = 1.85, CI [1.34–2.26], *p* <0.001) were statistically significant risk factors for abdominal wall endometriosis. Other variables that were significantly associated with abdominal wall location of endometriosis foci in the previous correlation analysis were not identified as significant risk factors (Table 4).

## 4. Discussion

As the causality of endometriosis in the studied patients cannot be linked entirely to the cesarean section intervention, the significant correlation and risk factor analysis place c-sections as an important independent factor associated with this condition. Our research connects the iatrogenic mechanism of the endometrial cell spread to the peritoneal cavity and abdominal wall, and it raises awareness of the need for careful management of surgical interventions involving the uterus. Although 42.2% of patients with a history of cesarean section were identified as having abdominal wall endometriosis, and although it is an insignificant percentage, it is worth mentioning that 5.4% of abdominal wall endometriosis was also found in patients with no history of cesarean section. This can be explained by endometrial seeding from other invasive procedures that were identified in this study or other causes that cannot be explained with the existing evidence presented by the current research. Until now, few previous studies and reviews have described cases of iatrogenic endometriosis developed after cesarean section [19], laparoscopic gastric bypass [22], and cholecystectomy [23]. In recent past years, clinicians have proposed several different possible theories and hypotheses for the occurrence of iatrogenic endometriosis: metaplasia, retrograde menstruation, venous or lymphatic metastasis, and mechanical transplantation. However, according to two reports, it is believed that mechanical transplantation is responsible for the evolution of iatrogenic endometriosis as the great majority of the cases of iatrogenic endometriosis occur after the incision of the gravid uterus [24]. In addition, in an old case report, it has been suggested that endometrium during pregnancy creates a favorable microenvironment with certain characteristics that make transplantation and implantation particularly successful [24].

Previous research describes the hypothesis of endometrial cells being present in the wound healing process, and these newly implanted endometrial cells can benefit when it comes to the source of nutrition and protective barrier provided by clot formation; growth can be sustained by the secretion of growth factor α by suppression of local immune response. Furthermore, the endometrial cell might already exist in the peritoneal cavity before surgical intervention. It can also appear from contamination from surgical mobilization, stretching, and tissue traction. Cesarean section can expose many endometrial cells during the surgery, which can be caught in the freshly manipulated abdominal wound. Additionally, a variable amount of amniotic fluid containing active endometrial cells might flood and invade the abdominal wound because the hysterotomy is performed during cesarean section. Cells can easily be transported into the pelvic floor through the amniotic fluid and are carried onto the skin, subcutaneous tissues, or muscles near the surgical incision [25]. The amniotic fluid may facilitate the disconnection of active cells [26]. The Pfannenstiel incision and the vertical midline incision are the most often utilized abdominal skin incisions. While the vertical midline incision facilitates abdominal entrance and results in less bleeding, it also increases the risk of incisional hernia and results in a less aesthetically attractive scar. On the other hand, the Pfannenstiel incision reduces the likelihood of incisional hernia and results in a more attractive appearance. However, the Pfannenstiel incision often requires more dissections and may result in more blood loss after dissection.

Even though, statistically speaking, scar endometriosis occurs most often after obstetric interventions; some studies suggest that women who have a record of laparoscopic supracervical hysterectomy might have a higher risk of advancing into scar endometriosis because the pelvic cavity can be exposed to endometrial cells, especially if morcellation is performed without a containment device [27]. It is also believed that pneumoperitoneum used in laparoscopic surgery was the major factor that contributed to tumor cell implantation [28,29]. Because pneumoperitoneum creates a pressure gradient with a subsequent outflow of gas, cells can float through the port wounds. This does not take place in routine surgery. Nevertheless, wall endometriosis after laparoscopic surgery is rare. Some studies suggest that obesity can offer a wide surgical surface for the entrapment of endometrial cells and may impact iatrogenic endometriosis and its severity [30].

Several limitations of the present study include the retrospective approach of studying endometriosis as an iatrogenic complication, presenting the need to correlate these rare findings with the surgical procedures involving the uterus that patients underwent. Moreover, a group of controls was not included to facilitate the statistical analysis, leaving the findings at an observational level.

## 5. Conclusions

Cesarean sections are an independent risk factor for abdominal wall seeding of endometrial tissue, which can be regarded as a form of iatrogenic endometriosis. Gynecologists should be suspicious of any woman who presents with discomfort at an incisional location, most often after pelvic surgery. Understanding the mechanisms of iatrogenic endometriosis will hopefully allow less invasive and more specific methods to be developed that facilitate prevention and early diagnosis. Although there is no definitive standard for diagnosing and managing atypical cases of iatrogenic endometriosis, it often involves thorough excision of the lesion and histological examination to confirm the diagnosis and rule out cancer. This research serves as a future direction for studies aiming to develop prevention mechanisms for endometrial cells seeding of the abdominal wall during cesarean sections.

## Figures and Tables

**Figure 1 ijerph-19-02791-f001:**
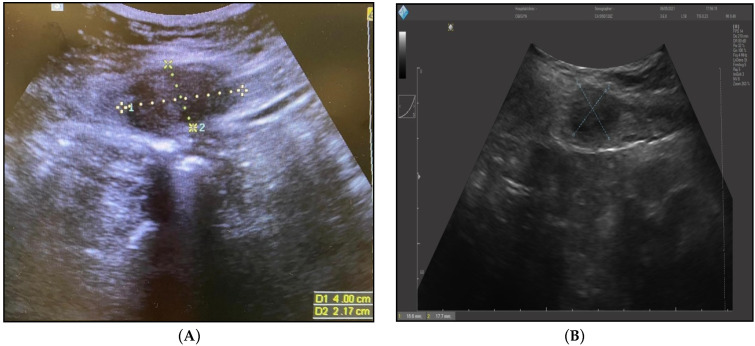
(**A**) Abdominal ultrasound of endometrioma after six months post-cesarean section (40/21.7 mm). (**B**) Abdominal ultrasound of endometrioma after three years post-cesarean section (18.6/17.7 mm).

**Figure 2 ijerph-19-02791-f002:**
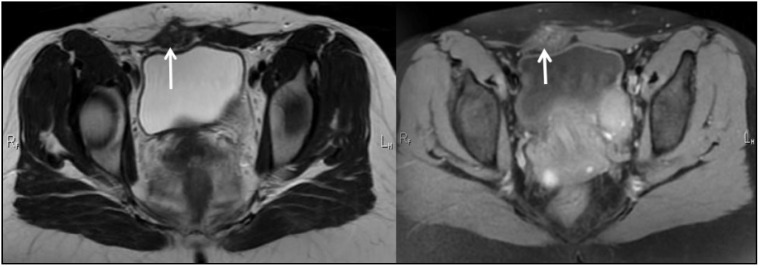
Pelvic MRI T1, T2: Deep Infiltrating Endometriosis, STAGE III, ENZIAN 3B, Abdomino-pelvic wall endometriosis (6-month postoperative aspect).

**Figure 3 ijerph-19-02791-f003:**
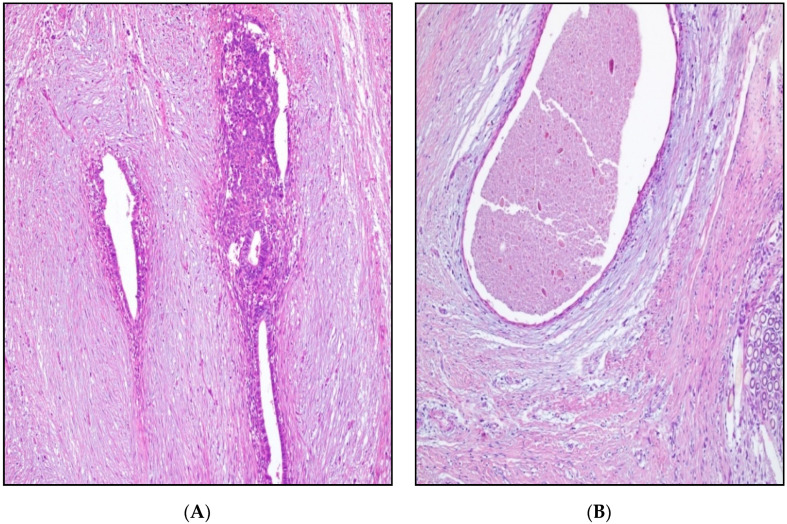
(**A**) Endometriosis in post-cesarean scar: endometrial glands, cytogenic chorion, and blood extravasations surrounded by young connective tissue; HEx100. (**B**) Endometriosis in post-cesarean scar: cystic dilated endometrial gland and reduced cytogenic chorion surrounded by loose myxoid and dense connective tissue; granuloma (bottom right); HEx100.

**Table 1 ijerph-19-02791-t001:** General characteristics of endometriosis cases in association with cesarean section history.

Characteristics	*n* = 71 (Frequency)	Mean ± SD
Patient age, years		33.2 ± 5.4
Age at surgical intervention		28.4 ± 7.2
Number of cesarean deliveries		
1	42 (59.2%)	
2	18 (25.4%)	
>2	11 (15.4%)	
Type of incision		
Pfannenstiel	62 (87.3%)	
Classic (midline)	9 (12.7%)	
Number of endometriomas		
1	14 (19.7%)	
>1	57 (80.3%)	
Location of the endometrioma		
Abdominal incision site	30 (42.2%)	
Distant from the incision site	41 (57.8%)	
Location of endometrioma in the abdominal wall		
Adipose layer	9 (12.7%)	
Fascia layer	40 (56.3%)	
Muscular layer	15 (21.1%)	
Peritoneum	4 (5.7%)	
Abdominal cavity	3 (4.2%)	
Duration between c-section and symptoms, months		27.5 ± 17.4
Duration between symptoms and treatment, months		30.2 ± 15.9
Symptoms		
Abdominal mass	44 (61.9%)	
Cyclic pain	62 (87.3%)	
Dysmenorrhea	49 (69.1%)	

SD—Standard Deviation.

**Table 2 ijerph-19-02791-t002:** Comparison between endometriosis study groups.

Variables *	History of C-Section (*n* = 71)	No History of C-Section (*n* = 55)	*p*-Value
Age, mean ± SD	33.2 ± 5.4	31.4 ± 6.4	0.089
Gestations			0.093
1	34 (47.9%)	37 (67.2%)	
2	23 (32.4%)	11 (20.0%)	
>2	14 (19.7%)	7 (12.8%)	
Pregnancies			0.050
1	39 (54.9%)	41 (74.5%)	
2	25 (35.2%)	9 (16.4%)	
>2	7 (9.9%)	5 (9.1%)	
Contraceptive use	22 (30.9%)	12 (21.8%)	0.250
Uterine minimally invasive procedures			
No procedures	37 (52.1%)	19 (34.5%)	0.049
Amniocentesis	3 (4.2%)	5 (9.1%)	0.266
Endometrial biopsy	6 (8.4%)	4 (7.2%)	0.808
Endometrial ablation	5 (7.0%)	2 (3.6%)	0.407
Uterine fibroid embolization	1 (1.4%)	2 (3.6%)	0.415
Hysteroscopy	9 (12.6%)	6 (10.9%)	0.391
Operative vaginal delivery	10 (14.1%)	8 (14.5%)	0.941
Curettage	18 (25.3%)	21 (38.2%)	0.122
Endometriosis foci features			
Size (mm), mean ± SD	32.2 ± 4.8	34.8 ± 5.5	0.005
Weight (g), mean ± SD	48.6 ± 7.0	53.1 ± 8.3	0.001
Endometriosis foci position			
Ovaries	22 (30.9%)	28 (50.9%)	0.023
Fallopian tubes	16 (22.5%)	17 (30.9%)	0.289
Uterosacral ligaments	7 (9.9%)	6 (10.9%)	0.847
Douglas pouch	7 (9.9%)	3 (5.4%)	0.364
Perimetrium	6 (8.4%)	2 (3.6%)	0.517
Rectum	3 (4.2%)	-	-
Vagina	3 (4.2%)	-	-
Abdominal wall	30 (42.2%)	3 (5.4%)	<0.001
Peritoneum	4 (5.7%)	5 (9.1%)	0.454
Intraoperative look			
No adhesions	38 (53.6%)	44 (80.0%)	0.002
Isolated adhesions	33 (46.4%)	11 (20.0%)	0.003
Isolated endometriosis	34 (47.9%)	39 (70.9%)	0.009
Multiple adhesions and/or endometriosis	37 (52.1%)	17 (29.1%)	0.017

* Data reported as *n* (frequency) unless specified differently; SD—Standard Deviation.

**Table 3 ijerph-19-02791-t003:** Correlation analysis for abdominal wall endometriosis.

		Invasive Procedures	Multiple Foci	Abdominal Wall Location	Foci Weight	Age	Foci Size	C-Section
Invasive procedures	Rho	1	0.465 **	0.240 *	−0.190	0.150	−0.069	0.316
	*p*-value		0.001	0.032	0.009	0.185	0.040	0.020
Multiple foci	Rho	0.465 **	1	0.469 **	−0.489 **	0.486 **	0.305 **	0.488 **
	*p*-value	0.001		0.001	0.001	0.001	0.001	0.001
Abdominal wall location	Rho	0.240 *	0.469 **	1	−0.315 **	0.171	0.036	0.523 **
	*p*-value	0.001	0.001		0.004	0.127	0.746	0.001
Foci weight	Rho	−0.190	−0.489 **	−0.315 **	1	0.499 **	−0.363 **	−0.194
	*p*-value	0.090	0.001	0.004		0.001	0.009	0.081
Age	Rho	0.149	0.486 **	0.171	0.499 **	1	0.229 *	0.124
	*p*-value	0.185	0.001	0.127	0.001		0.041	0.397
Foci size	Rho	−0.069	0.305 **	0.036	−0.363 **	0.229 *	1	−0.258
	*p*-value	0.040	0.005	0.746	0.009	0.041		0.003
C-section	Rho	0.316	0.488 **	0.523 **	−0.194	0.124	−0.258	1
	*p*-value	0.020	0.001	0.001	0.081	0.397	0.003	

** Correlation is significant at the 0.01 level (2-tailed); * Correlation is significant at the 0.05 level (2-tailed).

**Table 4 ijerph-19-02791-t004:** Risk factors associated with abdominal wall endometriosis.

Factors	OR	95% CI	*p*-Value
Age			
<35	1.04	0.61–1.25	0.462
≥35 ^	0.87	0.55–1.01	0.528
Number of gestations			
1 ^	1.09	0.80–1.24	0.409
>1	1.16	0.84–1.31	0.274
Number of pregnancies			
1 ^	1.07	0.94–1.01	0.230
>1	1.32	1.01–1.64	0.038
Foci size			
<32 mm	0.89	0.66–1.07	0.175
≥32 mm ^	0.92	0.81–1.05	0.097
Foci weight			
<48 g	0.94	0.72–1.24	0.229
≥48 g ^	0.98	0.79–1.03	0.206
Minimally invasive procedures			
Yes	1.27	1.12–1.58	0.063
No ^	1.08	0.92–1.17	0.273
C-section			
Yes	1.85	1.34–2.26	<0.001
No ^	1.16	1.08–1.33	0.162

OR—Odds Ratio; CI—Confidence Interval; ^—reference category.

## Data Availability

The data presented in this study are available on request from the corresponding author.

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
