# Peer review of "Challenges in Diagnosis and Prevention of Iatrogenic Endometriosis as a Long-Term Surgical Complication after C-Section"

_ijerph, 2022, doi:10.3390/ijerph19052791_

Round 1

Reviewer 1 Report

Interesting study reviewing patients with abdominla wall endometrioma. However, no additional informations are reported compare to the litterature.

Specific comments:

abstract: please focus on aims and provide more details on the material and methods and statistics.

Introduction: please foocus on rationale and aims and summarize.

Material and methods: more details on the procedures performed would be valuable.

Results: ok

Discussion: avoid the use of first person and verbiage. Remove "we think". Need to integrate more teh resulst of the study as currently, most of the discussion is not related to the findings of the study.

Conclusion: may be eedited to focus only on the results of this study.

Author Response

Dear reviewer,

We all appreciate your feedback and the time taken to evaluate our manuscript. Therefore, in order to improve our paper, we made the following edits based on your advice:

  1. The abstract section was extensively edited.
  2. In the introduction part we developed the aims of the study (lines 100-102).
  • Lines 76-77: we changed the wording so that reference 11 can follow our meaning and the symptoms described.
  • Lines 79-80 were modified to change the meaning for the care of iatrogenic endometriosis.
  • Please let us know if you consider some specific changes needed in this section.
  1. Materials and methods:
  • Added three more inclusion criteria that were not initially specified, in the attempt to control the existence of endometriosis before c-section (lines 115-117).
  • Added a sentence specifying how the diagnosis and procedures were codified (lines 126-128), and mentioned the procedures performed to explore the foci of endometriosis (line 137).
  1. Results:
  • Added a “c-section” variable in the correlation table that was previously omitted, and completed the correlation interpretation in the paragraph between lines 237 and 241.
  • Performed a multiple regression analysis to identify risk factors for abdominal wall endometriosis. Added Table 4 for this purpose, and the interpretation in the paragraph between lines 244 and 248.
  1. Discussion:
  • We removed the usage of first person “we” from the first paragraph (lines 252-254).
  • We explained the presence of endometriomas of the abdominal wall in 5% of patients who have never undergone surgery between lines 257-263.
  1. Conclusions:
  • We rearranged this section for a better description of our findings and further recommendations.

Please let us know if you have any other recommendations.

Best regards,

The authors

Reviewer 2 Report

This is an interesting paper dealing with endometriosis after cesarean section. However, in my opinion,  before being suitable for publication, it needs an extensive revision.

Summary

I would suggest a more structured abstract, giving more details on the design and results of the study and shortening the first part (etiology of endometriosis) and the last part (suggestions on the management).

Introduction

The definition of iatrogenic endometriosis is extremely important for the whole meaning of this paper and I could not find it in the reference provided by the authors (Rindos and Mansuria, 2017). Indeed, the topic of that review is "cesarean scar endometriosis" which is rather different from "iatrogenic endometriosis". The authors should provide another reference or give their own definition, stating that it is the authors' point of view.

The same is true for references 2 which, again, pertains to "cesarean scar endometriosis". 

Reference 3 is a review on the pathogenesis of endometriosis and, again, I could not find any reference to iatrogenic endometriosis, apart from a generic sentence on the effects of uterine tissue scarring during surgery.

A long part of the introduction is dedicated to the diagnosis of endometriosis and to a comparison between different imaging techniques. This seems a bit unrelated to the aim of the paper.

Reference 11 does not deal with symptoms of "iatrogenic endometriosis" but with symptoms of "endometriosis" in general. 

 I would disagree with the statement that "the care of iatrogenic endometriosis should be similar to that of non-iatrogenic endometriosis". Indeed, an endometriosis implant in a cesarean scar can be easily removed and surgery would be the first approach; on the contrary, this is not true for non-iatrogenic implants inside the peritoneal cavity, which may be treated with medical therapy as first line treatment.

Materials and methods

The study was authorized in 2021 and is a retrospective study, therefore I understand that the authors revised the clinical files of the patients admitted between January 2010 and November 2021. This is not immediately clear and these paragraph should be rephrased.

I understand   that the patients were recruited among those who underwent surgery for pelvic pain and resulted to be affected by endometriosis after surgery. Is this correct?

What about patients with cesarean scar endometriosis and no pelvic pain??

Which were the inclusion criteria for patients without previous cesarean section? Had they never had any kind of abdominal surgery?

Only 55 patients with endometriosis and no previous cesarean section underwent surgery in 10 years in a large hospital such us Timisoara University Clinic of Obstetrics and Gynecology Bega (Spitalul Clinic de Obstetrică și Ginecologie Bega)? Were there other exclusion criteria?

The whole paragraph from line 115 to line 122 seems to suggest that the surgical procedures were performed during the study (i.e. after 2021). On the contrary they had already been performed. So they were performed according to the hospital protocols and not according to a study protocol. This should be made clear. Moreover, the description of the methods used for the pathological exam are redundant.

Results

Line 131: some words are missing.

In table 1 the units for measuring time should be stated (years and months?)

The images of ultrasound and MRI and those of the histology are redundant. The scope of this paper is not to discuss the diagnosis of endometriosis.

In table 2 in "intraoperative look"  a patient is missing in the first column (38+32=70 and not 71).

A total of 7 patients had undergone endometrial ablation, 15 hysteroscopy and  and 40  uterine curettage. Was this before or after the appearance of pelvic pain?

In table 1 60% of patients had an endometrioma at the incision site. However, in table 2 only 42% of patients in the c-section group had an endometrioma of the abdominal wall. How is this possible?

Discussion

Again, it seems to me that this is a study on endometriosis after cesarean section and not on "iatrogenic endometriosis".

The authors should explain the presence of endometriomas of the abdominal wall in 5% of patients who have never undergone surgery.

Author Response

Dear reviewer,

We all appreciate your helpful feedback and the time taken to evaluate our manuscript. Indeed, your understanding of our manuscript is thorough and compelling. In order to improve our paper, we made the following edits based on your advice and the other reviewers’ comments:

  1. The abstract section was extensively edited in order to give it a better structure by focusing more on results.

Introduction

  1. We reshaped the definition of iatrogenic endometriosis in order to fit our ideas, and we changed the first reference title (lines 53-54).
  2. According to our new definition for iatrogenic endometriosis, references numbered 2 and 3 should be properly integrated in the text.
  3. In the introduction part we developed the aims of the study (lines 100-102). Please let us know if you consider some specific changes needed in this section.
  4. Lines 76-77: we changed the wording so that reference 11 can follow our meaning and the symptoms described.
  5. Lines 79-80 were changed based on your advice on the care of iatrogenic endometriosis.

Materials

  1. Lines 109-110: We specified that patients were admitted to our clinic between those dates.
  2. Lines 115-117: We added three more inclusion criteria that were not initially specified, in the attempt to control the existence of endometriosis before c-section.
  3. Lines 126-128: We added a sentence specifying how the diagnosis and procedures were codified (lines 126-128), and mentioned the procedures performed to explore the foci of endometriosis (line 137).
  4. Indeed, 55 patients is a small number for patients with endometriosis. Even though there were many more patients with endometriosis and no previous caesarean section, we included only the cases that had a pelvic laparoscopic surgical exploration or laparotomy specifically for endometriosis.
  5. Lines 115-122 (now lines 138-140): we clarified how the procedures were performed before this study, and according to existent hospital protocols

Results

  1. Line 131 (now line 154): we edited the sentence with missing words.
  2. Added the units of measurement in Table 1.
  3. We considered keeping those images since they facilitate the understanding of atypical locations of endometriosis. However, we can remove them if you believe they are truly useless.
  4. Table 2 “intraoperative look”: we corrected the missing patient.
  5. The 7 patients that had undergone those invasive procedures before having pelvic pain.
  6. Indeed, there was a typo when creating the tables. Only 42% of patients in the c-section group had an endometrioma of the abdominal wall. We corrected table 1 accordingly.
  7. Added a “c-section” variable in the correlation table that was previously omitted, and completed the correlation interpretation in the paragraph between lines 237 and 241.
  8. Performed a multiple regression analysis to identify risk factors for abdominal wall endometriosis. Added Table 4 for this purpose, and the interpretation in the paragraph between lines 244 and 248.

Discussion

  1. This is true, we consider endometrial tissue seeding after c-section a type of iatrogenic endometriosis.
  2. We explained the presence of endometriomas of the abdominal wall in 5% of patients who have never undergone surgery between lines 257-263.
  3. We rearranged the conclusions section for a better description of our findings and further recommendations.

Please let us know if you have any other recommendations.

Best regards,

The authors

Reviewer 3 Report

I have two point to highlight.

1. Describe more in depth the control arm choice of cases.

2. I don't understand why using linear correlation. I suggest to remake calculation by using logistic regression OR odds ratio calculation. Results would be more soundness and easy to understand for readers.

From other point of view, I have not anything to amend.

Good work available for meta-analyses.

Author Response

Dear reviewer,

We all appreciate your feedback and the time taken to evaluate our manuscript. Therefore, in order to improve our paper, we made the following edits based on your advice and the other reviewers’ recommendations:

  1. The abstract section was extensively edited.
  2. In the introduction part we developed the aims of the study (lines 100-102).
  • Lines 76-77: we changed the wording so that reference 11 can follow our meaning and the symptoms described.
  • Lines 79-80 were modified to change the meaning for the care of iatrogenic endometriosis.
  1. Materials and methods:
  • Added three more inclusion criteria that were not initially specified, in the attempt to control the existence of endometriosis before c-section (lines 115-117) and to clarify the study arms.
  • Added a sentence specifying how the diagnosis and procedures were codified (lines 126-128), and mentioned the procedures performed to explore the foci of endometriosis (line 137).
  1. Results:
  • Added a “c-section” variable in the correlation table that was previously omitted, and completed the correlation interpretation in the paragraph between lines 237 and 241.
  • Performed a multiple regression analysis to identify risk factors for abdominal wall endometriosis. Added Table 4 for this purpose, and the interpretation in the paragraph between lines 244 and 248.
  1. Discussion:
  • We explained the presence of endometriomas of the abdominal wall in 5% of patients who have never undergone surgery between lines 257-263.
  1. Conclusions:
  • We rearranged this section for a better description of our findings and further recommendations.

Please let us know if you have any other recommendations.

Best regards,

The authors

Round 2

Reviewer 1 Report

The authors adequately addressed the comments from reviewers.

Author Response

Dear reviewer,

We appreciate your positive feedback and thank you for the productive collaboration.

Best regards, 

The authors

Reviewer 2 Report

I think the paper is now suitable for publication

Author Response

(The authors gave the same response as above.)

Reviewer 3 Report

I am satisfied by Authors' answers, except for Table 4. For intelligibility, readers have to know the reference categories.

Odds ratio of 1.85 for Cesarean section means that, in case of previous Cesareans, risk of endometriosis increases. but what about other variablers? specifically, which is the reference category for previous pregnancies?

I recommend to clarify this item before publication.

Your,

U.

Author Response

Dear reviewer,

We appreciate your positive feedback.

Thank you for noticing the missing reference part in Table 4. We made the appropriate corrections to include a reference category for all variables in the table. For example, the reference category for previous pregnancies was 1. Please see the attached revised manuscript.

Best regards,

The authors